# Students' Knowledge of Climate Change, Mitigation and Adaptation in the Context of Constructive Hope

Ilkka Ratinen 

Faculty of Education, University of Lapland, 96101 Rovaniemi, Finland; ilkka.ratinen@ulapland.fi

**Abstract:** Humanity is living in a climate emergency where climate change should be significantly mitigated; additionally, greater efforts should be made to adapt to it. To date, relatively little research has been carried out on young people's skills in terms of them mitigating and, in particular, adapting to the changes caused by climate change. The complex climate change issues of mitigation and adaptation are conceptually difficult for children because climate change is not directly evidenced in their daily lives. This study focuses on looking at mitigation and adaptation from the perspective of children. Meaning-making coping strategies enable the maintenance of constructive hope regarding climate change. In the present study, elementary and secondary students ($n$ = 950) responded to an online questionnaire. Statistical methods were used to gather data on how students' general knowledge of climate change and their mitigation and adaptation knowledge predicted their constructive hope regarding climate change. This study reveals that the students had a relatively high level of constructive hope and that general climate change knowledge predicted students' constructive hope well.

**Keywords:** climate change; knowledge; mitigation; adaptation; constructive hope



## 1. Introduction

Even though climate change is a difficult issue to understand, young people are an important group to incorporate into the efforts to combat climate change. Climate strikes for young people inspired by Greta Thunberg have spread all over the world, and they have had a remarkable influence on the current climate law in Finland [1]. The ability to mitigate and adapt to climate change requires knowledge of mitigation measures and adaptation strategies. Mitigation aims to stabilize and reduce the levels of greenhouse gas (GHG) in the atmosphere [2]. However, due to the GHG concentration that is already present in the atmosphere, some effects of climate change will continue despite mitigation efforts, and therefore adaptation is needed. Adaptation means the adjustment of natural or human systems in response to actual or expected climatic stimuli or their effects, which moderates harm or exploits beneficial opportunities [2].

Current research has failed to fully recognize how children and young people understand climate change mitigation and adaptation. Studies reveal, for instance, preservice teachers' challenges in understanding climate change mitigation [3] and the difficulties in incorporating mitigation and adaptation strategies into elementary school instruction [4]. However, we do know that constructive hope strongly predicts climate change mitigation knowledge [5] and that problem-focused and meaning-focused coping are positively associated with late adolescent proenvironmental behavior [6].

Evidence is still lacking on how students' knowledge about the causes of climate change and their knowledge of mitigation and adaptation strategies are associated with constructive hope. Li and Monroe [7] examined the role of wider climate knowledge being associated with hope. The psychology of hope, developed by Snyder [8], defines a positive sense of hope as a force for action. Three themes have been identified as important for constructive hope. Firstly, the trust in one's own ability to make a difference [9]. Secondly,

different individual and collective proenvironmental behaviors are effective in trying to reach the subgoals [8], and thirdly, concrete actions so that it seems as if hope is evoked by the actions themselves [5]. This study focuses on the latter theme in order to determine how students' general knowledge of climate change and their knowledge related to climate change mitigation and adaptation predict their levels of constructive hope.

In Finland, geography is a key school subject where climate change is generally taught [10] and, in this study, we aim to discuss the content of such geography lessons as it relates to climate change. Based on earlier research, we know that environmental measures are connected in schools and in textbooks with small and easy to carry out actions; that is, the discourse of small acts [11]. Moreover, there is also some evidence that concrete solution-oriented tasks increase students' hope regarding climate change [4]. However, more research is needed to clarify climate change mitigation and adaptation knowledge among young people. In the media, social media and even in presidential talks, young climate activists have been belittled regarding their knowledge on the subject, and their ability to criticize climate policy has been questioned. Notwithstanding, the youth are a logical target group for climate change education because they will be the ones handling the future negative consequences of the climate problem. They are also consumers and citizens, representing a part of the climate change problem due to their lifestyle. Therefore, it is vital to understand this group's relationship with constructive hope and to explore how knowledge regarding climate action promotes their hope for the future.

## 2. Connectiveness between Climate Change Knowledge, Mitigation and Adaptation and Constructive Hope

Knowledge of climate change can increase hopelessness, which leads to pessimistic attitudes towards the future and ultimately does not help in mitigating climate change [6,7,12,13]. Therefore, teaching really matters regarding how to keep hope up. Ojala and Bengtsson [6] examined how young people's positivity was positively correlated with problem-focused and meaning-focused coping, while negativity was associated with de-emphasizing climate-related problems. Knowledge of climate change mitigation and adaptation processes helps to maintain problem-focused coping strategies because then people know what they can do. Similarly, knowledge helps in choosing meaningful strategies. An incorrect scientific understanding of climate change can lead to denialism or to those without proper knowledge about climate change being misled towards an alternative truth. Fischer [14] examined the abovementioned issues from the perspective of the philosophy of science: Social constructivism seems to help us better understand the rise of post-truth, because knowledge constructs are always, in part, artefacts of a particular societal system and its culture. A recent debate among climate change deniers reveals that they have come to more or less accept the high level of agreement about the human contribution to global warming, but they have shifted to arguing that the problem is too big to be able to mitigate against or adapt to response skepticism [14,15]. Capstick and Pidgeon [16] found that adult concerns about climate change are significantly more pronounced for response skepticism ($r = 0.40$). Özdem, Dal, Öztürk, Sönmez and Alper [17] pointed out that only a minority of students think that climate change can be mitigated. Pupils' epistemic perceptions of climate change are not well known, but it is known that an erroneous conceptual understanding is difficult to correct [18]. Therefore, it is important that the mechanism of climate change is learnt scientifically and accurately. If $CO_2$ is not understood as the cause of climate change, it is also difficult to understand mitigation strategies relating to limiting $CO_2$ emissions.

What are the key messages that could lead students towards a correct understanding of climate change mechanisms? First, education should lead learners towards replacing their incorrect mental models so as to understand climate change [18]. Many researchers have repeated that students mainly have two nonscientific mental models on the subject. One focuses on GHGs and the other on the ozone layer, as well as there being variations in their understanding of those models. The first model aligns with the scientifically accurate model that a defined GHG layer exists in the atmosphere, yet the impacts of climate change

are not understood as a change in the radiation balance [18–21]. Students can have a model that focuses on the role of GHGs, but that model can be incomplete or inaccurate [18].

The second model inaccurately connects climate change to the ozone layer [18,21–24]. The dominant model illustrates that the presence of GHGs or other air pollutants produces holes in the ozone layer. These "concrete" holes allow more heat and/or UV rays to reach the Earth's surface and thus increase the temperature of the planet. It is noteworthy that, in terms of climate change education, if people incorrectly understand the mechanism of climate change, then they cannot choose effective mitigation procedures, or they experience denial-based hope for the future climate [5]. Moreover, it is obvious that these two nonscientific mental models are very similar for students from elementary school up to university level [18,24,25].

The way in which young people understand mitigation and adaptation has been studied much less frequently. According to Aitken, Chapman and McClure [26], mitigating climate change is recognized as an increasingly urgent task that requires understanding a range of different strategies, including voluntary behavior change. The perceived risk arising from climate change and the perception that humans influence climate change seem to be the strongest predictors of mitigation action. Secondary school students, as Özdem et al. [17] have shown, may place the responsibility for climate change mitigation on individuals, national governments and environmental and international organizations. However, Pettersson [27] has pointed out that children and adolescents, just as Myllyniemi [28] also found, often place the responsibility for climate change on other people but not on themselves. International agreements, legislation and education have been considered effective mitigation strategies by students in some studies [17,29]. Bofferding and Kloser [30] found that adolescents demonstrated a limited understanding of adaptive responses to climate change. Hermans and Korhonen [31] pointed out that, although secondary school students experienced climate change as a risk and they thought climate change mitigation to be relevant, their own willingness to act was quite limited.

In terms of students' individual climate change mitigation actions, a similar trend was noted. Truelove and Parks [32] found that college students underestimated the mitigating potential of adjusting thermostats and reducing meat consumption but overestimated the impact of littering causing climate change. Hermans and Korhonen [31] found that secondary school students were most willing to switch off lights and electrical appliances to mitigate climate change. About half of them were willing to sell things in second-hand shops and to cycle or walk moderate distances. They were least willing to buy things from second-hand shops and to refrain from using motor-driven vehicles, such as mopeds. Secondary school students have been shown to be willing to act to mitigate climate change personally only if minor inconveniences, minor costs, or minor lifestyle changes are implied. When planning climate education in schools, it should be kept in mind that students are less willing to undertake actions that might be seen as involving greater personal effort, such as reducing meat consumption, using public rather than private transport, or buying fewer fashion items [29,33].

In recent years, as our knowledge has increased, young people's worry and pessimistic views on our global future have increased, not least when it comes to environmental problems [34–37], and those emotions have a negative impact on constructive hope. Pihkala [38] pointed out that ecoanxiety also manifests itself as "practical anxiety", which leads to the gathering of new information and a reassessment of behavior options. In climate change education, it is important to maintain a perspective of hope, as hopelessness leads to pessimistic attitudes towards the future and, ultimately, this does not help in mitigating climate change [6,12,39–41].

In climate change education, however, it is important to distinguish between optimism and hope. Eagleton [42] defines hope as emphasizing the meaningfulness of creating a certain attitude, whereas optimism refers more to faith in success. This is particularly important with regard to climate change, as there are no definite guarantees that we will succeed in mitigating and adapting to climate change [7]. Orr points out [43] that

individuals sometimes have a tenet in optimism that achieving a better future is relatively easy. However, this kind of future will be very difficult to achieve. For some, hope is a virtue, a habit of finding meaningfulness in life and of resilience, of not giving up. This aspect could be opened up via constructive hope.

This study examines youth climate change knowledge and its mitigation and adaptation connections to constructive hope. There is evidence that constructive hope seems to relate to proenvironmental behavior [5,9,40,44]. Ojala [44] calls this ability to face environmental risks and uncertainty, a belief that one's own actions and the actions of others can make a difference and of finding positive meaning in action, constructive hope. Additionally, Ojala and Bengtsson's [6] results can be applied in constructive climate education, as problem-focused and meaning-focused coping strategies were found to be positively related to environmental considerations. In recent years, there has been a derogatory debate about the climate movement of young people, where young people are thought not to know enough about climate change. The debate has focused on climate anxiety among young people and it is necessary to open up the debate from the perspective of their constructive hope [38]. Examining the connection between knowledge and constructive hope will shed further light on the link between hope and optimism.

The knowledge about climate change and its mitigation and adaptation related to constructive hope can be explained by Li and Monroe's study [7]. They found that when young people felt that they and others could address problems effectively, they were more likely to feel hope ($r = 0.76$). Many young people claim that they feel hopeful because there are things that they themselves can do to fight climate change (problem-focused coping $r = 0.59$ and meaning-focused coping $r = 0.21$) [6]. Moreover, adolescents are more likely to show both problem-focused and meaning-focused coping when parents and friends respond in solution-oriented and supportive ways [6]. Trott [45] found that children's sense of agency was a confluence of hope, confidence and the motivation to affect change, and its source was children's climate change awareness and actions. According to Ojala [37], adolescents are more likely to express constructive hope regarding climate change when their own environmental engagement is relatively high ($r = 0.22$) and when their teachers respect their emotions and offer support ($r = 0.42$).

## 3. Aims of the Study

Taking the earlier studies as a whole, there is still inconsistency in the results on students' views on climate change mitigation and adaptation. In particular, we have a large gap in understanding constructive hope as a way to find the means to transform our society towards a sustainable future [5,46].

Although we know about students' perceptions and their feelings of hopelessness regarding climate change, there are still shortcomings about how their knowledge of climate change mitigation and adaptation strategies can be used to measure the level of constructive hope. Successful climate change education needs to concentrate on climate actions that can truly decrease the amount of GHGs in the atmosphere. The present study seeks to answer the following research questions:

- How much do students know about climate change and the relevant mitigative and adaptative strategies?
- How does their knowledge predict their constructive hope around the climate change issue?

## 4. Methods

### 4.1. Procedure and Participation

Nine hundred and fifty elementary and secondary students living in Finland participated in the study. The average age was 13.6, and the sample included 49% of girls and 51% of boys. Both elementary classes and secondary classes were included in the study (10% were fifth-graders, 20% were sixth-graders, 15% were seventh-graders, 31% were eighth-graders, and 24% were ninth-graders). However, the sample should be considered as a convenience sample, because we cannot report an exact response rate. The students

answered an online questionnaire at their school and were guaranteed anonymity. The students had the opportunity to refuse to take part in the research and for those that took part, the parents' consent forms were signed. Teachers conducted the survey, and they were instructed to guide students in responding to the survey following the instructions in the research form. The students' response times ranged from 30 to 45 min. The sample is the same as in [5] but this study focuses on detailed climate change knowledge instead of overall hope regarding climate change.

### 4.2. Measures and Statistical Tests

This study explored the relationship between climate change knowledge and constructive hope and the knowledge questions included 30 items in total. The Intergovernmental Panel on Climate Change [2] defines mitigation as a human intervention to reduce the sources or enhance the GHG sinks and the adaptation of the process of adjustment to align with the actual or expected climate and its effects. These definitions provided the basis for the questionnaire statements.

To measure the students' climate change knowledge (Table 1), the participants were asked to respond to the question, "Climate change is because . . . " and 13 items were used: "There is too much greenhouse gas", "There is intensive forest logging", "Nitrogen oxides are released from fertilizers", "There is too much IR radiation that stays on the Earth", "Purchased products generate greenhouse gases", "Landfills produce methane", "There is too much consumption of milk and dairy products", "Emitted IR radiation is absorbed by greenhouse gases", "We use fossil fuels such as oil and coal", "the Earth has ozone zone", "We are using nuclear power", "We are using wind power" and "Factories and cars increase temperature". To develop the statements, the study by Ratinen [18] was used. The Cronbach's alpha for the items was 0.86. The items were followed by alternative answers: Strongly disagree = 1, Disagree = 2, Neither disagree nor agree = 3, Agree = 4 and Strongly agree = 5.

**Table 1.** PCA for climate change knowledge.

| Climate Change Increases Because | Component Loadings | | | Item Scores | |
|---|---|---|---|---|---|
| | Knowledge I | Knowledge II | Knowledge III | M (SD) | Agree % |
| there is too much greenhouse gas (GHG) | 0.75 | | | 3.92 (1.07) | 69.7 |
| too much IR radiation stays on the Earth | 0.75 | | | 3.71 (1.05) | 49.7 |
| we use fossil fuels such as oil and coal | 0.72 | | | 3.96 (1.10) | 68.2 |
| purchased products generate GHGs | 0.70 | | | 3.45 (1.07) | 49.9 |
| nitrogen oxides are released from fertilizers | 0.59 | | | 3.46 (1.03) | 66.5 |
| landfills produce methane | 0.52 | | | 3.34 (0.97) | 40.2 |
| there is intensive forest logging | 0.48 | | | 3.70 (1.18) | 60.9 |
| we are using wind power | | 0.85 | | 3.21 (1.56) | 49.2 |
| we are using nuclear power | | 0.62 | | 3.65 (1.16) | 57.3 |
| there is too much consumption of milk and dairy products | | 0.57 | | 3.75 (1.32) | 64.0 |
| the Earth has an ozone hole | | | 0.78 | 3.19 (1.14) | 42.6 |
| emitted IR radiation is absorbed by GHGs | | | 0.65 | 3.10 (0.98) | 28.2 |
| factories and cars increase temperature | | | 0.55 | 3.33 (1.19) | 46.7 |
| Eigenvalue | 5.003 | 1.384 | 1.002 | | |
| Total variance explained (%) | 45.0 | 10.6 | 7.7 | | |

Note: Agree = Agree and Strongly agree.

The students' knowledge of climate change mitigation (Table 2) was captured by the question, "If the climate is to change as little as possible, we need to . . . " and was measured with ten items: "Use public transport more", "Walk and bicycle more", "Avoid food waste", "Eat local food", "Reduce dairy products", "Favour a vegetarian diet", "Reduce the purchase of goods", "Lower the room temperature", "Travel inland" and "Use trains not planes". The Cronbach's alpha for the climate change knowledge mitigation items was 0.81. The items were followed by alternative answers: Strongly disagree = 1, Disagree = 2, Neither disagree nor agree = 3, Agree = 4 and Strongly agree = 5.

**Table 2.** PCA for climate change mitigation knowledge.

| If the Climate Is to Change as Little as Possible, We Need to | Component Loading | | | Item Scores | |
| --- | --- | --- | --- | --- | --- |
| | Mitigation I | Mitigation II | Mitigation III | M (SD) | Agree % |
| use public transport more | 0.78 | | | 3.83 (1.19) | 66.2 |
| walk and bicycle more | 0.76 | | | 4.23 (1.07) | 80.2 |
| avoid food waste | 0.75 | | | 3.94 (1.06) | 66.6 |
| eat local food | 0.71 | | | 3.89 (1.16) | 67.6 |
| reducing dairy products | | 0.80 | | 2.75 (1.25) | 27.5 |
| favour a vegetarian diet | | 0.75 | | 3.24 (1.31) | 45.1 |
| purchase less goods | | 0.64 | | 3.25 (1.19) | 44.5 |
| by lowering room temperature | | 0.63 | | 3.01 (1.12) | 30.1 |
| travel inland | | 0.59 | | 3.08 (1.27) | 37.5 |
| use trains not planes | | | 0.96 | 2.59 (1.22) | 19.4 |
| | | | | | M = 58.2 |
| Eigenvalue | 4.152 | 1.068 | 1.028 | | |
| Total variance explained (%) | 41.5 | 10.7 | 10.3 | | |

Note: Agree = Agree and Strongly agree.

The students' knowledge of climate change adaptation (Table 3) was captured by the question, "When we adapt to climate change we ... " and it was measured with eight items: "Need to strengthen the river banks", "Need to strengthen the urban drainage network due to heavy rainfall", "May need to adapt to a longer, snowless period", "Have to use more pesticides", "Need to care more about the health of the elderly", "May have to shovel snow in April and May", "Need to reduce more slippery roads (sanding)" and "Have to clean more blue-green algae from beaches". The study by Bofferding and Kloser [30] was used to develop the statements. They stated that mitigation and adaptation might be complementary. Thus, the statements in this study illustrate how adaptive climate actions are needed if mitigation has not been adequate. The Cronbach's alpha for the climate change adaptation knowledge items was 0.78. The items were followed by alternative answers: Strongly disagree = 1, Disagree = 2, Neither disagree nor agree = 3, Agree = 4 and Strongly agree = 5.

**Table 3.** PCA for climate change adaptation knowledge.

| When We Adapt to Climate Change We | Component Loading | | Item Scores | |
| --- | --- | --- | --- | --- |
| | Adaptation I | Adaptation II | M (SD) | Agree % |
| need to strengthen the river banks | 0.75 | | 3.30 (0.97) | 37.7 |
| need to strengthen the urban drainage network due to heavy rainfall | 0.75 | | 3.36 (0.97) | 41.5 |
| may need to adapt to a longer, snowless period | 0.73 | | 3.64 (1.10) | 57.3 |
| have to use more pesticides | 0.55 | | 2.96 (0.96) | 24.3 |
| need to care more about the health of the elderly | 0.51 | | 3.83 (0.98) | 65.7 |
| may have to shovel snow in April and May | | 0.79 | 3.34 (1.09) | 45.1 |
| need to reduce more slippery roads (sanding) | | 0.78 | 3.29 (1.08) | 41.1 |
| have to clean more blue-green algae from beaches | | 0.53 | 3.64 (1.03) | 56.6 |
| | | | | M = 49.3 |
| Eigenvalue | 3.207 | 1.034 | | |
| Total variance explained (%) | 40.1 | 13.0 | | |

Note: Agree = Agree and Strongly agree.

Finally, the students' attitudes towards constructive hope regarding climate change was measured (Table 4). Ojala's study [9] was used for the construction of the statements. Ratinen and Uusiautti [5] revealed two PCAs for climate change hope: constructive and hope based on denial. The constructive hope for used this study. The students expressed their opinions by answering the question, "I think about climate change with hope because ... ". In total, seven items were used: "We have to take responsibility for climate issues and take those issues seriously", "I can change my behaviour and, together, we can have a positive impact on the climate", "Awareness has increased", "I can help to reduce climate change in many ways", "People in environmental organizations can solve climate change", "There is the Paris Climate Agreement" and "Science and technology promote

solutions to climate change". The Cronbach's alpha for the constructive hope items was 0.87. The items were followed by alternative responses regarding agreement: Not at all = 0, Only slightly = 1, Slightly = 2, Neither slightly nor a lot = 3, Quite a lot = 4, A lot = 5, Very much so = 6.

**Table 4.** PCA for constructive hope regarding climate change.

| I Think about Climate Change with Hope Because | Component Loading Constructive Hope | Item Scores M (SD) | A Lot % |
|---|---|---|---|
| we have to take responsibility for climate issues and take those issues seriously | 0.81 | 4.03 (1.55) | 40.9 |
| I can change my behaviour and, together, we can have a positive impact on the climate | 0.81 | 3.96 (1.48) | 36.5 |
| awareness has increased | 0.78 | 3.90 (1.45) | 37.1 |
| people in environmental organizations can solve climate change | 0.77 | 3.58 (1.48) | 26.4 |
| I can help to reduce climate change in many ways | 0.75 | 3.63 (1.56) | 30.0 |
| there is the Paris Climate Agreement | 0.67 | 3.39 (1.39) | 20.7 |
| science and technology promote solutions to climate change | 0.64 | 3.29 (1.39) | 17.2 |
| Eigenvalue | 3.934 | | |
| Total variance explained (%) | 56.2 | | |

Note: A lot = A lot and Very much so.

A principal component analysis (PCA) was used for the calculation of principal scores with the regression method for variables. The scores were also used for a hierarchical regression analysis (Table 5) to predict relations between climate change-related knowledge and constructive hope. The regression analysis [47] has proven to be a good method in environmental psychometric studies. The statements about climate change knowledge, mitigation and adaptation knowledge and constructive hope regarding climate change were further analysed statistically for gender differences using *t*-tests. Age groups were compared using a Kruskal–Wallis analysis of variance (KW-ANOVA), because Levene's test indicated that there was a lack of homogeneity in the variances. The students were also asked for their own opinions on their own climate change knowledge and their trust in themselves being able to adapt to climate change. The idea for this was based on Ojala's [44] definition of constructive hope: to believe that one's own actions can make a difference. The statements were: "I can explain the causes and consequences of climate change" and "I trust that people can adapt to climate change". The items were followed by alternative answers: Strongly disagree = 1, Disagree = 2, Neither disagree nor agree = 3, Agree = 4 and Strongly agree = 5. As all the variables were not normally distributed, nonparametric Spearman's rhos were calculated.

**Table 5.** Hierarchical regression model predicting constructive hope related to climate change.

| | Constructive Hope | | |
|---|---|---|---|
| | Step 1 β | Step 2 β | Step 3 β |
| Climate change knowledge I | 0.372 *** | 0.229 *** | 0.108 ** |
| Climate change knowledge II | 0.115 *** | - | - |
| Climate change knowledge III | 0.114 *** | - | - |
| Adaptation knowledge I | | 0.288 *** | 0.244 *** |
| Adaptation knowledge II | | | 0.187 *** |
| Mitigation knowledge I | | | 0.199 *** |
| Mitigation knowledge II | | 0.211 *** | 0.090 ** |
| Mitigation knowledge III | | | - |
| $R^2$ | 0.165 *** | 0.263 *** | 0.289 *** |
| Adjusted $R^2$ | 0.162 *** | 0.259 *** | 0.283 *** |
| $\Delta R$ | 0.165 *** | 0.098 *** | 0.023 *** |

*** $p \leq 0.001$; ** $p < 0.005$.

## 5. Results

As Table 1 shows, the students' knowledge about climate change was moderate but variable. On average, most students (55.3%) responded correctly to the statements. The students seemed to understand the causes of climate change better than the GHG-emitted IR radiation aspect (M = 3.10). In particular, fossil fuels and GHGs as accelerators of climate change reached a relatively high level of agreement among the students. It should be noted, however, that statements represented the common scientific understanding of the causes of climate change, but less than 70% of the students agreed with the statements. There were also misunderstandings. Students quite commonly agreed that nuclear (57.3%) and wind power (49.2%) causing climate change and they can confuse heat from cars and factories to the radiative force (46.7%). The PCA was a suitable method for calculating the principal scores, as the Kaiser–Meyer–Olkin (KMO) measure for sampling adequacy (KMO) was 0.89. This three principal component solution explained 56.8% of the total variance and the factor loadings were satisfactory (0.50 or greater, not logging), (see Table 1). Knowledge I represents scientifically correct knowledge and knowledge II associates with the uncertain understanding of causes of climate change. The knowledge II is not very clear, and students probably associate nuclear power and wind energy with energy production altogether, which naturally increases climate emissions. Knowledge III indicates misunderstanding the mechanism of climate change. The girls' climate change knowledge I was better than that of the boys, t(948) = 7.353, $p < 0.000$, d = 0.45. Age did not statistically significantly affect climate change knowledge. The students' confidence in their own climate change knowledge was weakly correlated with their actual climate change knowledge ($r_s = 0.218$, $p < 0.01$).

The students' knowledge of climate change mitigation also varied substantially (Table 2). On average, students correctly responded 58.2% of the time to the statements. They agreed that walking and biking were the most important strategies for climate change mitigation (M = 4.23). Instead, according to the students, changing from planes to trains was not a significant way to reduce climate change (M = 2.59). It is a fact that avoiding dairy products, lowering the room temperature and travelling inland instead of travelling abroad help to solve climate change, but less than 40% of the students thought that way. The PCA was used for the estimation of students' opinions on their climate change mitigation knowledge. A KMO of 0.88 showed that the data was suitable for PCA testing with varimax as the rotation method. This principal component solution accounted for 62.5% of the total variance and the factor loadings were satisfactory (0.50 or greater) (see Table 2). Finally, three scales were created: Mitigation knowledge I, II and III. Mitigation I represents everyday issues in the child's and young person's everyday life that they consider relevant for mitigating climate change. Mitigation II consists of variables that also indicate everyday issues but that are outside the control of the child and the young people because their parents make purchases. Mitigation III only includes a single variable related to modes of travel. The single variable for the last component is strong because it can explain the deviation in the observed variables as well as the five variables of the second component. Girls' climate change mitigation knowledge was better than that of boys. They knew more about mitigation I t(948) = 7.001, $p < 0.000$, d = 0.45 and mitigation II t(947.3) = 7.524, $p < 0.000$, d = 0.45. However, boys' knowledge of mitigation III was better than that of girls t(948) = −3.171, $p < 0.002$, d = 0.21. Age only statistically significantly affected mitigation III knowledge, with both 11-year-olds and 16-year-olds knowing it was best to use a train instead of a plane to help to mitigate climate change $\chi^2(6) = 34.616$, $p < 0.000$. The students' confidence in their own climate change knowledge correlated slightly with their climate change mitigation I scores ($r_s = 0.124$, $p < 0.01$) and the correlations were weak for mitigation II ($r_s = 0.079$, $p < 0.05$) and mitigation III ($r_s = −0.091$, $p < 0.01$).

The students' knowledge about climate change adaptation was lower than their general climate change knowledge and their knowledge regarding mitigation (Table 3). On average, students correctly responded 49.3% of the time to the statements. The students had difficulty understanding that, as rainfall increases, flood adaptation is

needed (M = 3.30 and 3.36), and as temperatures rise, pesticides are needed to control pests (M = 2.96). The latter result is contradictory, because as the temperature rises, more attention must also be paid to caring for the elderly (M = 3.83). The PCA was used for the estimation of the students' opinions on their climate change adaptation knowledge. A KMO of 0.84 showed that the data was suitable for a PCA with varimax as the rotation method. This two principal component solution explained 53.1% of the total variance and the factor loadings were satisfactory (0.50 or greater) (see Table 3). Finally, two scales were created: Adaptation knowledge I and II. Adaptation I represents general adaptation measures to climate change. However, the factor loading for elderly people and the use of pesticides remained relatively low. Adaptation II consists of variables that indicate the need for cleaning snow in late spring, removing algae from beaches and reducing slippery roads. The girls' climate change adaptation knowledge was better than that of the boys: for adaptation I t(948) = 2.685, *p* < 0.007, d = 0.17 and adaptation II t(948) = 5.551, *p* < 0.000, d = 0.36. Age only statistically significantly affected adaptation I and 14-year-olds knew the most $\chi^2(6)$ = 15.972, *p* < 0.003. Their confidence in their own climate change knowledge was weakly correlated with their climate change adaptation knowledge ($r_s$ = 0.136, *p* < 0.01).

Table 4 indicates students' relatively high levels of constructive hope, but it varied quite remarkably (SDs). Climate change strategies being their responsibility (M = 4.03), students' beliefs that they could change their behavior together with others (M = 3.96) and an increased awareness (M = 3.90) mostly represented the students' constructive hope. Instead, it would seem that it is not very important for students that climate agreements (M = 3.39) and science and technology (M = 3.29) could increase their constructive hope. Similarly, only 30% of the students believed that they had the means to mitigate climate change (M = 3.63). A PCA was also used to measure the levels of hope in the students regarding climate change. A KMO of 0.90 showed that the data was suitable for a PCA, with varimax as the rotation method. This one principal component solution accounted for 56.2% of the total variance and the eigenvalue was 3.934. The factor loadings were satisfactory (.50 or greater). The girls' constructive hope was higher than that of the boys, t(948) = 4.001, *p* < 0.000 d = 0.26. Age did not statistically significantly affect their constructive hope regarding climate change. The students' confidence in their own climate change knowledge was slightly correlated ($r_s$ = 0.208, *p* < 0.01), as was their trust in their ability to adapt ($r_s$ = 0.158, *p* < 0.01), with their constructive hope regarding climate change.

As depicted in Table 5, that scientifically correct climate change knowledge was a significant predictor in the hierarchical regression model (β.372 ***, *p* < 0.001). In Step 2, misunderstanding of climate change lost predictivity and adaptation was entered in the model, and adaptation I (i.e., adaptations due to changing weather patterns) predicted constructive hope well (β.288 ***, *p* < 0.001). In Step 3 of the model, the other control variables were entered. In this final step, knowledge of climate change adaptation I (β.244 ***, *p* < 0.001) and mitigation I (β.199 ***, *p* < 0.001) were the strongest positive predictors. The results indicated quite a strong connection between climate change knowledge and constructive hope. Consequently, it seems to be evident that having scientifically correct climate change knowledge (e.g., $CO_2$ causes climate change, walking and bicycling are mitigative strategies and taking care of old people is an adaptive consequence) do indeed strongly predict students' constructive hope. On the other hand, the result also suggests that misconceptions about the causes of climate change may to some extent support students' constructive hope regarding climate change (β.114 ***, *p* < 0.001). However, this effect will cease with the addition of the right knowledge about mitigation and adaptation of climate change. The result suggests that solution-oriented climate change education can increase hope. The full model accounted for 29% of the variance, F(3, 941) = 11.2, *p* < 0.000.

## 6. Discussion

This study shows that students understood incompletely the issues around climate change and its mitigation and adaptation. Their understanding of the climate change mechanism as one of a changing radiation balance was inadequate and they confuse

climate change and ozone depletion causally together. This result aligns with previous studies [18,19]. However, the students had internalized the effect of GHGs on climate change quite well. This result is significant from the perspective of climate change education. Understanding climate change supports the choice of effective mitigation measures. Once GHGs are known to cause climate change, it will be easier to find ways to cut such emissions. Trott [48] found that children felt empowered by their knowledge and that they did take action to minimize climate harm. In this study, the students seemed to understand mitigation as more valid when the strategies are able to be undertaken in their daily lives and when they have opportunities to bring some influence via mitigative acts. This result is similar to that from Hermans and Korhonen's [31] and Bofferding and Kloser's [30] studies, where only a few students recognized material production and consumption as a possible form of mitigation. In this study, when mitigation measures were more related to consumption limitation and they were outside of the children's everyday control, their knowledge was more limited. The students' willingness to prevent climate change by using less private transport was examined in a large study [49]. The study revealed that about two-thirds of the students believed that greater use of public rather than private transport would reduce climate change. Surprisingly, choosing a train instead of a plane was not, in the students' view, a significant mitigative climate change measure. This result differs from the finding by Bofferding and Kloser [30], where the mode of transportation was the most popular mitigation strategy.

The students had less of an understanding of climate change adaptation than of climate change mitigation, and the result is similar to that in the study by Bofferding and Kloser [30]. Based on the results, climate change adaptation was seen as adaptation to higher temperatures rather than as adaptation to increased rainfall. The result is not surprising, as there has been less talk of adaptation in school and society. In particular, one must first understand what climate change causes in order to understand the resulting changes in daily life. On the other hand, climate change often appears as a rising temperature in the media, for instance [50]. However, adaptation cannot be ignored in climate change education. Its importance will increase in the future, as climate change will likely cause severe and sudden weather events [2], and because students may confuse mitigation and adaptation strategies, climate change education may benefit from increased activities that address adaptations to climate change [30,51].

Constructive hope promotes meaning-making coping strategies for climate change mitigation and adaptation [9,44,52]. Ojala [44] concluded that meaning-focused coping seems to be an especially constructive strategy, since it is positively associated with environmental efficacy and engagement, as well as well-being. Based on this study, increasing awareness of climate change, shared responsibility, and the belief that we can mitigate climate change by changing our behavior seem to increase students' constructive hope. However, regarding climate change education, it is interesting that students do not believe in science and technology or in the Paris Climate Agreement, at least in terms of this belief increasing their constructive hope. Therefore, the students' trust in legislation is lower than, for example, that found in the study by Özdem et al. [17]. Instead, they thought that on a personal level, they had to consider climate change seriously and take responsibility, but their own possibilities regarding being able to reduce climate change varied substantially. The present study promotes Ojala's [44] idea of constructive hope because the students' confidence in their own climate change knowledge was weakly correlated—as was their trust in adaptation—with their constructive hope regarding climate change.

The present study reveals that students holding accurate knowledge about climate change and its mitigation and adaptation does predict their constructive hope regarding climate change quite well. Ojala [37] found that trust and optimism are important aspects of constructive hope and that young people who experience a high level of constructive hope indicate that they are willing to vote for a party that works for sustainability issues. In addition, this study highlights the importance of focusing on constructive hope in climate change education. The environmental psychology research argues that the appearance

of some kind of hopefulness did not always prove useful for climate change mitigation, as individuals needed a more realistic view, such as energy efficacy, where their own responsibility was sufficiently recognized [6,47]. The present study promotes this idea because the students mostly agreed with the statement related to their everyday life. Li and Monroe [7,39] found that when young people feel concern about environmental problems and believe that they and others can address these problems effectively, they are more likely to feel hope. The results of the present study are in line with the aforementioned research.

### 6.1. Implications for Instruction

Kagawa and Selby [53] pointed out that three dimensions of climate change education—understanding and attentiveness, mitigation and adaptation—correspond to key elements in climate risk reduction education. In summary, the study leans on the abovementioned idea and presents noteworthy aspects that can be considered in climate change education in geography lessons but also in other lessons on climate change as well. In conclusion, the following points should be emphasized in climate change education to maintain constructive hope:

1. The cause of climate change due to the increasing amount of anthropogenic GHGs in the atmosphere needs to be paid attention to when teaching.
2. $CO_2$ is the most significant anthropogenic GHG, and attention should be paid to its reduction, as students may have difficulty combining the origin of $CO_2$ with various mitigation measures.
3. Focusing on the adaptations required due to climate change is also needed, because there are clear gaps in understanding. Adaptation will be needed in the future, even if climate change mitigation is successful, because GHGs (especially $CO_2$) are long acting in the atmosphere.
4. Taking responsibility for climate issues and noticing the learners' skills and their beliefs in terms of collaboration should be emphasized in the teaching. Solution-oriented mitigation and adaptation strategies should be highlighted because the right knowledge significantly supports constructive hope.

When young people are faced with complicated and difficult mitigative and adaptive strategies, it is important to support their constructive hope. This challenging task will probably be more successful when students are focusing on information that is personally relevant by relating it to local and everyday issues. Confidence in authorities and science could be managed by connecting young people with scientists and activists who can share their work and stories, by supporting them in projects to care for the climate in their schools and communities and by engaging them through experiential inquiry-based [18] and arts-based methods [45]. In their review article, Monroe et al. [54] identified four aspects of effective climate change education: (1) engaging students in deliberative discussions, (2) interacting with scientists within their learning tasks, (3) addressing their misconceptions relating to climate change and (4) implementing school or community projects into instruction. This study supports the results of the previous review study both directly and indirectly. In conclusion, this study points out the insufficient climate change knowledge of the students, but deliberative dialogic discussion [18] and interactions with scientists appear to help motivate students to learn more [55] and community projects seem to engage students in climate change education [45]. Knowledge has an important role in maintaining constructive hope, yet it can be a trigger for practical anxiety [38]. However, this anxiety should be carefully considered in climate change education and an empirical study on this issue is needed.

### 6.2. Limitations of the Study

This study explores a new area of research with some of the scales being created specifically for this study. Thus, although the reliability of the scales was satisfactory, in future studies they should be further validated. Caution should be exercised in interpreting the responses, as the effect sizes, or values that represent the magnitude of the differences,



remained quite small for all variables. More research is needed to assess whether some educational strategies are more effective for students who do not yet have opportunities to learn about practices or experiences in climate change mitigation and adaptation. Additionally, the correlations were of medium strength. The results of this study are based on a convenience sample of Finnish elementary and secondary students. To be able to generalize the results in a broader way, it is important to use random sampling in future studies and to include young people from different cultures and countries [49].

**Funding:** The data collection of this research was funded by the Finnish Climate Change Panel.

**Institutional Review Board Statement:** Ethical review and approval were waived for this study because none of the criteria for ethical review and approval defined by the Finnish Advisory Board on Research Integrity (TENK) were met in the research design. In Finland, ethical review in human sciences applies only to precisely defined research configurations.

**Informed Consent Statement:** Informed consent was obtained from all subjects involved in the study.

**Data Availability Statement:** The data are not publicly available due to privacy protection of participants.

**Conflicts of Interest:** The authors declare no conflict of interest.

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
