# Peer review of "Students’ Knowledge of Climate Change, Mitigation and Adaptation in the Context of Constructive Hope"

_education, doi:10.3390/educsci11030103_

Round 1

Reviewer 1 Report

Congratulations to authors: a solid article, well-built.

Author Response

I edited paper and used template is given.

Reviewer 2 Report

COMMENTS:

Comment 1

Please, prepare references according to Template file. Chapter name Literature rename to References! It is not possible to see references [4], [5], [18], [25], in chapter References!

Comment 2:

Chapters in paper is necessary to numerate! Please, see Template file!

FINAL OPINIONS:

Please, prepare the paper in detail according to instructions for authors and Template file! Organization of paper is very weak! Methodology is not clear! Results are not clearly presented!

Reconsider after major revision!

Author Response

1) Organization of paper

I used the template is given and added the references.

2) Methodology is not clear

I erased "analytical". The papers like this utilizing statistical analysis usually do not present mathematical equations.  Here I referred e.g. Vainio et al. study. If I have understood correctly equations are presented in thesis and and the papers where are statistical analysis somehow further developed. 

3) Results are not clearly presented

I tried to present the results in the tables as concisely as possible and to save space. Therefore, I did not make the figures of the knowledge, for example. When I red the accepted paper on Education Science Journal I noticed that this seems to be common in the journal. However, I recognise that If I would have studied, for example, learning by using pre- and post-test the figure would have added value. I also presented the regression analysis in the table. If I would have used e.g. the SEM-model then it would have been useful to present a figure. I hope this is clear explanation. 

Reviewer 3 Report

The paper is well written and the statistics indicate a solid review of the data.  I could not find an error in the statistics and data set.

The References are all recent with the oldest only from 2000.

In the Reference section, the paper has a citation on page #1 for references #4 and #5.  On the reference list, there are no citations for #4 and #5, only the numeral.  The same is tru for citation #25.  Only the numeral and no reference is listed.  These should either by added or removed.

Overall the paper is very interesting and thoughtful.  I believe it is timely and adds to the discussion about Climate control.

Author Response

The references are added and template for used. 

Round 2

Reviewer 2 Report

COMMENTS:

Comment 1

Line 225:

Wrong: 3.1. Procedure and Participation

Correct: 4.1. Procedure and Participation

Comment 2:

Line 241:

Wrong: 3.2. Measures and Statistical Tests

Correct: 4.2. Measures and Statistical Tests

Comment 3:

The spaces between individual titles, tables and text are not uniform (see chapters 5 and 6, Table 2, Table 3 and Table 5). Also, spacing between references in chapter References is necessary to adjust!

FINAL OPINIONS:

According to comments, it is clear that organization of paper is better, but still weak! Please, try to fix it!

Accept after minor revision!

Author Response

Thank you! The template I used is strange. The spaces varied. But I adjusted the spaces exactly for headings and all tables. Also the style of references has been done by template. I do not know does the 1st line hanging use for reference list. Maybe editor know that.